# Optimizing LED Light Intensity and Photoperiod to Promote Growth and Rooting of Medicinal Cannabis in Photoautotrophic Micropropagation

**DOI:** 10.3390/biology14060706

**Published:** 2025-06-16

**Authors:** Juwen Liang, Fang Ji, Qing Zhou, Dongxian He

**Affiliations:** 1College of Water Resources and Civil Engineering, China Agricultural University, Beijing 100083, China; liangjuwen@cau.edu.cn (J.L.); jifang@cau.edu.cn (F.J.); cauzhou@cau.edu.cn (Q.Z.); 2Key Laboratory of Agricultural Engineering in Structure and Environment, Agricultural Engineering in Structure and Environment, Ministry of Agriculture and Rural Affair, Beijing 100083, China

**Keywords:** *Cannabis sativa* L., continuous lighting, photoautotrophic micropropagation, plantlets, net CO_2_ exchange amount

## Abstract

*Cannabis sativa* L. is a crop of significant medicinal and economic value, widely used in the fiber, food, and medical fields. With the increasing demand for medical cannabis products, the production of healthy, uniform, and disease-free seedlings has become increasingly important. However, conventional micropropagation methods often face challenges, such as slow growth, low rooting rates, and susceptibility to contamination, which limit their suitability for large-scale cultivation. In this study, we employed photoautotrophic micropropagation under aseptic, sugar-free conditions, regulating LED light intensity and photoperiod to promote the growth and rooting of cannabis plantlets. Our results demonstrate that a light intensity of 100–150 µmol m^−2^ s^−1^ combined with a 20 h d^−1^ photoperiod significantly promoted root elongation, increased leaf development, and enhanced overall quality of plantlets. In contrast, continuous lighting and excessively high light intensities induced light stress responses, negatively affecting plantlet growth. Thus, this study provides data support for optimizing the light environment in the photoautotrophic micropropagation of sterile, high-quality cannabis plantlets, thereby facilitating the broader application of cannabis in the medical and pharmaceutical fields.

## 1. Introduction

Cannabis (*Cannabis sativa* L.) is an annual herbaceous plant from the Cannabaceae family, widely used for fiber, food, and medicinal purposes for thousands of years. With the advancement of scientific research on the extraction of active compounds and the evaluation of its medicinal value, alongside the relaxation of cannabis cultivation policies in several countries, more than 70 nations have now approved its use in the medical field [1]. According to the latest report by Grand View Research, the global cannabidiol market size was valued at USD 9.14 billion in 2024 and is expected to expand at a compound annual growth rate of 15.60% from 2025 to 2030 (https://www.grandviewresearch.com/industry-analysis/cannabidiol-cbd-market (accessed on 14 April 2025)). The continuous increase in medical demand has not only driven the expansion of medicinal cannabis cultivation but also significantly increased the demand for disease-free, growth-synchronized, and genetically uniform female cannabis seedlings as starting materials for cultivation [2,3].

Cannabis can be propagated through seed propagation, tissue culture, and vegetative propagation. While seed propagation is cost effective, the resulting offspring often exhibit phenotypic variation, leading to genetic instability and reduced consistency in the final product [4]. Although conventional tissue culture can provide sterile and genetically uniform materials, it still faces several challenges. For instance, callus is highly susceptible to vitrification with poor redifferentiation ability. Moreover, contamination issues are prominent, plantlets grow slowly, have low rooting success rates, and transplant survival rates are often unsatisfactory [5,6,7]. In contrast, vegetative propagation ensures genetic homogeneity and stable seedling growth, providing uniform planting materials for large-scale cultivation, thus making it the preferred method for medicinal cannabis propagation [4,8]. Cutting propagation, a common vegetative propagation method, however, requires substantial cultivation space to maintain mother plants, typically occupying 10–15% of the total cultivation area [4]. Moreover, as the cultivation period extends, cuttings face heightened vulnerability to pests and diseases, including *Botrytis cinerea* and *Golovinomyces cichoracearum* [9,10], as well as *Phorodon cannabis* [11]. These pests and diseases inhibit plant growth and compromise the quality of the final product. This issue is particularly concerning in medicinal cannabis cultivation, as it can negatively affect the accumulation of bioactive compounds [12,13].

Micropropagation, as an effective alternative to cutting propagation, employs tissue culture techniques to cultivate plants in a controlled, sterile environment. This method has been proven to minimize the spread of pests and diseases while improving propagation efficiency [14,15,16]. Recent studies have explored various explants for cannabis micropropagation, including stem segments, cotyledons, leaves, meristems, and flowers [4,7,17,18,19,20,21]. However, these methods rely on the use of sealed containers and sugar-containing media, resulting in high production costs, low rooting success rates, and poor graft survival. To address these challenges, Professor Toyoki Kozai from Chiba University in Japan proposed the concept of photoautotrophic micropropagation (PAM) in the late 1980s [22]. PAM involves cultivating explants under sterile conditions on a sugar-free culture medium without organic components such as vitamins and plant growth regulators, utilizing ventilated containers to enhance photosynthesis and transpiration, thereby significantly promoting plantlet growth and development [23]. For instance, Kodym and Leeb [24] proposed a PAM protocol suitable for rooting shoot tips of cannabis. This method involves cultivating shoot tips in forced-ventilation containers with rockwool as supporting material to promote growth. Subsequently, repeatedly harvesting shoot tips from plantlets and placing them on rockwool for rooting induction achieved a rooting rate of up to 97.5% within 3–4 weeks. Similarly, Lubell-Brand et al. [14] and Zarei et al. [25] proposed similar cannabis propagation protocols, where shoot proliferation is initially carried out on agar, followed by rooting induction with rockwool, achieving rooting rates of 75–100%. These studies primarily focused on the effects of culture medium composition, explant type and length, and basal stem injury on growth and rooting under PAM conditions.

During cannabis micropropagation, the photoperiod is typically maintained between 16 and 24 h d^−1^, with a light intensity of 50–100 μmol m^−2^ s^−1^ to minimize transpiration loss during adventitious root formation [26,27,28]. Zarei et al. [25] found that under the photoperiod of 18 h d^−1^ and light intensity of 50 μmol m^−2^ s^−1^, 2.5 cm stem segments successfully rooted within six weeks. Moreover, their study on the effect of light intensity on rooting revealed that the light intensity of 150 μmol m^−2^ s^−1^ significantly improved the rooting success rate of cuttings compared to treatments with 50 and 100 μmol m^−2^ s^−1^ [2]. Murphy and Adelberg [29] investigated the effect of light intensity ranging from 25 to 167 μmol m^−2^ s^−1^ on the rooting of cannabis shoot tips. Their results indicated that at the light intensity of 120 μmol m^−2^ s^−1^, the rooting success rate of explants approached 100%. Moher et al. [30] reported that LED light quality had no significant effect on rooting success, but combinations of blue light and UV radiation significantly improved overall seedling quality. Similarly, Wannida et al. [31] found that blue light had no significant impact on root induction, whereas far-red light supplementation effectively promoted rooting without inducing excessive stem elongation. Consistently, McKay et al. [32] showed that the addition of 5% far-red light significantly increased shoot number and shoot length in an in vitro multiple-harvest system for cannabis plantlet production. These findings suggest that optimizing the lighting environment for explant is a crucial factor in improving rooting rates and seedling quality [33,34]. Therefore, this study aims to explore the optimal combination of light intensity and photoperiod suitable for the PAM of medicinal cannabis, with the goal of improving the overall quality of plantlets.

## 2. Materials and Methods

### 2.1. Plant Materials and Culture Conditions

The plant materials included two medicinal cannabis cultivars developed by Dutch Passion (Netherlands): a short-day cultivar (CBD Charlotte’s Angel, Charlotte) and a day-neutral cultivar (CBD Auto Charlotte’s Angel, Auto Charlotte). Initially, seeds with full grains and no damage were selected. After rinsing with tap water, the seeds were soaked for 24 h, then placed in petri dishes containing sterile filter paper to induce germination. Sterile pure water was sprayed twice daily to keep the filter paper moist. After 3–4 d, germinated seeds were transferred to sponge blocks (L23 mm × W23 mm × H23 mm) within seedling trays (L330 mm × W240 mm × H50 mm) filled with nutrient solution. The nutrient solution had an electrical conductivity (EC) value of 1.0 ± 0.2 mS cm^−1^ and pH of 5.8 ± 0.2. This solution refers to the research results of Bernstein’s team [35] and is composed of the following compounds: Ca(NO_3_)_2_·4H_2_O, KNO_3_, NH_4_H_2_PO_4_, (NH_4_)_2_SO_4_, K_2_SO_4_, MgSO_4_·7H_2_O, DTPA-Fe, Na_2_B_4_O_7_·10H_2_O, MnSO_4_·H_2_O, ZnSO_4_·7H_2_O, (NH_4_)_6_Mo_7_O_24_·4H_2_O, CuSO_4_·5H_2_O, with concentrations of 865.7, 316.4, 111.3, 12.5, 118.1, 618.6, 40.0, 0.44, 2.50, 0.70, 0.04, 0.16 mg L^−1^, respectively. At 14 d after sowing, uniform seedlings with well-developed root systems were selected and transplanted into a hydroponic system. The system comprised two ABS cultivation beds (L1200 mm × W600 mm × H70 mm) and two ABS boards (4 mm thick) with planting holes (20 mm in diameter), with each layer having a height of 70 cm (Appendix A). Each cultivation bed transplanted 14 plants, resulting in a planting density of 19.4 plants m^−2^. The seedlings were grown under lighting conditions provided by white LED lamps (W6000K-18 W, R:B ratio of 1.0, Beijing Lighting Valley Technology Co., Ltd., Beijing, China), with the canopy PPFD of 300 μmol m^−2^ s^−1^ and a photoperiod of 18 h d^−1^. The same nutrient solution as the seedling stage was used, adjusted to an EC value of 1.8 ± 0.2 mS cm^−1^ and pH of 5.8 ± 0.2. In the controlled environment, temperature and relative humidity were maintained at 22 ± 1 °C and 60% ± 5% during the light period, and 18 ± 1 °C and 70% ± 5% during the dark period. CO_2_ concentration was not controlled.

Culture vessels were Magenta boxes (GA-7, L75 mm × W75 mm × H100 mm) with a volume of 380 mL. Each box was equipped with a lid containing two circular holes (8 mm in diameter), sealed with 18 mm polymer breathable membranes (Milliseal, pore size 0.47 μm, Japan) to facilitate gas exchange. The supporting material was a mixture of peat, vermiculite, and perlite in a 1:1:1 volume ratio. Each GA-7 contained 9.0 ± 0.5 g of this mixture, with a dry bulk density of approximately 0.15 g cm^−3^ and a volume of about 60 cm^3^. Additionally, 45 mL of nutrient solution containing 0.5 mg L^−1^ IBA was added to each GA-7. The solution composition was identical to that used in the seedling stage, with an EC value of approximately 1.8 mS cm^−1^ and pH of 5.8 ± 0.2. After preparation, the GA-7 vessels were sterilized in an autoclave at 121 °C for 20 min and subsequently stored in a sterile environment for later use.

Shoot tips (Appendix A) were collected from two-month-old female mother plants that were disease free and exhibited uniform growth. Each shoot tip had one to two fully expanded leaves, with an average length of 3.3 ± 0.3 cm, a stem diameter of 1.9 ± 0.2 mm, a fresh weight of 0.29 ± 0.08 g, and a dry weight of 0.06 ± 0.02 g. Explants were sterilized in 0.5% NaClO for 2 min, then rinsed three times with sterile water. Subsequently, under a sterile laminar flow cabinet, the base of each explant was trimmed at a 45° angle and then inserted into the supporting material in the GA-7, with one explant per vessel (Appendix A). The inoculated GA-7 were placed on culture shelves (L1200 mm × W600 mm × H1500 mm, with 6 layers and a layer height of 250 mm) in a controlled environment room. Temperature and relative humidity were maintained at 22 ± 1 °C and 60% ± 5% during the light period, and 18 ± 1 °C and 70% ± 5% during the dark period. CO_2_ concentration was not controlled (Appendix A).

### 2.2. Experimental Design

The experiment involved four light intensities (50, 100, 150, and 200 µmol m^−2^ s^−1^), three photoperiods (16, 20, and 24 h d^−1^), and two medicinal cannabis cultivars (Charlotte and Auto Charlotte), resulting in 24 treatments. Each treatment included 12 GA-7 and was replicated three times (Appendix A). White LED lamps (W4000K-18 W, R:B ratio of 1.8, Beijing Lighting Valley Technology Co., Ltd., Beijing, China) served as the artificial lighting source and were installed 15 cm above the GA-7. Light intensities were controlled by varying the number of LED lamps and moving the position of the vessels. Measurements were taken 15 cm below the LEDs using a portable light meter (LI-250A, LI-COR Inc., Lincoln, NE, USA), with specific values detailed in Table 1.

### 2.3. Measurements

#### 2.3.1. Growth Indicators

After 28 d of PAM, six uniform and healthy plantlets were randomly selected from each treatment for morphological and biomass analysis. Plant height and root length were measured using a ruler, while stem diameter was determined with a vernier caliper. The number of nodes and leaves per plantlet was manually counted. Each leaf was digitally scanned using a CanoScan LiDE400 (Canon Inc., Tokyo, Japan), and its area was analyzed using the Image Analysis function in Adobe Photoshop CC 2019 (Adobe Systems Incorporated, San Jose, CA, USA). Roots were gently washed to remove residual substrate. The fresh weight of roots and shoots was measured using an electronic balance (FA1204B, BioonGroup, Shanghai, China). The samples were then sterilized in an oven at 105 °C for 1 h, followed by drying at 75 °C for 72 h to obtain dry weight. Water content and Dickson’s quality index (DQI) were calculated according to Equation (1) and Equation (2), respectively.(1)$$Water\;content\;(\%)=\frac{Fresh\;weight-Dry\;weight}{Fresh\;weight}\times 100$$



(2)
$$DQI=Total\;dry\;weight/(\frac{Plant\;height\;(mm)}{Stem\;diameter\;(mm)}+\frac{Shoot\;dry\;weight}{Root\;dry\;weight})$$



#### 2.3.2. Photosynthetic Pigment Content

The chlorophyll content of the second leaf of the plantlets was determined following the method described by Li [36]. A 0.1 g leaf sample was excised, cut into small pieces, and immersed in 10 mL of 95% ethanol for 48 h in the dark. The absorbance of the extract was measured at 665, 649, and 470 nm using a spectrophotometer (UV3150, Shimadzu Corporation, Kyoto, Japan) to calculate the contents of chlorophyll a, chlorophyll b, and carotenoids, respectively.

#### 2.3.3. Antioxidant Enzyme Activity and Root Activity

Superoxide dismutase (SOD) activity was determined using the nitroblue tetrazolium photoreduction method [37], peroxidase (POD) activity by the guaiacol method [38], and catalase (CAT) activity by the ultraviolet absorption method [39]. Root activity was determined using the triphenyltetrazolium chloride reduction method [40].

#### 2.3.4. Net CO_2_ Exchange Amount

The CO_2_ concentration and temperature–humidity were measured using CO_2_ sensor (DCO2-TF series, Dihui Technology Co., Ltd., Beijing, China, Appendix A) and a split-type temperature and humidity sensor (DWS-T5W1-E-S1D, Dihui Technology Co., Ltd., Beijing, China, Appendix A). These sensors were integrated with LabVIEW 2021 software (National Instruments, Austin, TX, USA) for data communication, acquisition, processing, and storage. The DCO2-TF series CO_2_ sensors have a measurement range of 0–10,000 μmol mol^−1^ and can operate stably in high-humidity environments. In the experiment, four CO_2_ sensors and one temperature and humidity sensor were used to monitor CO_2_ concentration and temperature–humidity conditions inside the GA-7 under varying light intensity with 20 h d^−1^ photoperiod. Additionally, one CO_2_ sensor and one temperature and humidity sensor were placed in the culture room to monitor the ambient CO_2_ concentration and temperature–humidity conditions. To ensure accurate monitoring, two holes approximately 5 mm in diameter were drilled in the lids of the GA-7. The CO_2_ and temperature–humidity sensors were placed inside the L050-P20 vessels, with the signal wire holes sealed using Parafilm to prevent air leakage (Appendix A). The same setup was applied to the other three treatments (L100-P20, L150-P20, L200-P20), with one CO_2_ sensor placed inside each vessel, recording data once per min. Before the experiment and at weekly intervals during the experiment, all sensors were placed in a standardized reference environment to record CO_2_ concentration, temperature, and humidity over a 2 h period. Average values for each parameter were calculated, and experimental data were corrected by comparing differences between these average values, ensuring data accuracy. The air exchange rate of the GA-7 was measured using CO_2_ as a tracer gas according to the method described by Kozai et al. [41], resulting in a value of 4.5 h^−1^. Based on previous studies [42,43,44], the net CO_2_ exchange amount (NCEA, μmol vessel^−1^) for a given period (from t1 to t2) was calculated using Equation (3). The NCEA was calculated separately for the light and dark periods. The cumulative NCEA during the light and dark periods over the entire cultivation cycle were then summed to obtain the total net CO_2_ exchange amount in light (Light NCEA) and dark (Dark NCEA), respectively. Finally, Light NCEA and Dark NCEA were summed to calculate the total net CO_2_ exchange amount (Total NCEA) for the entire cultivation cycle.(3)$$NCEA=k\cdot \left\{N\cdot V\cdot \int_{t_{1}}^{t_{2}}\left(C_{out}\left(t\right)-C_{i1}\left(t\right)\right)\cdot dt-V\cdot \left(C_{i2}-C_{i1}\right)\right\}$$
where:k: Conversion factor from volume to molecular weight of CO_2_ (41.9 mol m^−3^ at 18 °C, 41.3 mol m^−3^ at 22 °C);N: The air exchange rate per min of the GA-7 (min^−1^).V: The ventilation volume of the GA-7 (m^3^);C_out_: The CO_2_ concentration in the culture room (μmol mol^−1^);C_i1_: The CO_2_ concentration inside the GA-7 at time t1 (μmol mol^−1^);C_i2_: The CO_2_ concentration inside the GA-7 at time t2 (μmol mol^−1^).

#### 2.3.5. Heatmap Analysis

The measured parameters, including plant height, root length, number of leaves, leaf area, fresh and dry weight of roots, stems, and leaves, root shoot ratio, DQI, photosynthetic pigment content, root activity, and antioxidant enzyme activity were normalized using the Z-score method. The normalization was centered on the mean value of the entire dataset to minimize differences between parameters. Subsequently, the normalized data were visualized as a clustered heatmap using the Heatmap function on the Chiplot website (Sweden, www.chiplot.online (accessed on 14 April 2025)).

### 2.4. Statistical Analysis

The experimental data were presented as “mean ± standard deviation” (n = 4). Differences among groups were analyzed using Duncan’s multiple range test (α = 0.05) in SPSS software (v26.0, IBM Corp., Armonk, NY, USA). Graphs were created using Prism 9.0 software (v9.4, GraphPad Software, San Diego, CA, USA, www.graphpad.com). Identical letters indicate no significant difference between groups, whereas different letters indicate significant differences. “NS” represents no significant effect. Additionally, biomass, antioxidant enzyme activity, root activity, and net CO_2_ exchange amount were fitted using a second-order polynomial regression model (Y = A + B × X + C × X^2^).

## 3. Results

### 3.1. Growth Performance

Light intensity and photoperiod significantly influenced the growth and rooting of plantlets (Figure 1). Under the same light intensity, both cultivars exhibited optimal growth and well-developed root systems, with a photoperiod of 20 h d^−1^. In contrast, under the same photoperiod, increasing light intensity initially enhanced the growth performance of both cultivars, but further increases led to inhibition, suggesting that excessive light intensity may induce photoinhibition. Furthermore, under a photoperiod of 16 h d^−1^, the leaves of plantlets maintained a healthy green color. However, under continuous lighting treatments, significant chlorosis was observed, indicating that excessively long photoperiod may cause nutrient deficiency or photosynthetic imbalance.

### 3.2. Growth Parameters

The light intensity and photoperiod significantly influenced the morphological growth parameters of plantlets (Table 2). Under the same photoperiod, plant height, root length, leaf number, leaf area, and DQI of both cultivars initially increased and then declined with increasing light intensity, reaching their highest values at 100 or 150 µmol m^−2^ s^−1^. Under the same light intensity, as the photoperiod increased, the root length, stem diameter, water content, and root shoot ratio of both cultivars exhibited a decreasing trend. The best performance was observed under 16 or 20 h d^−1^ photoperiods, whereas continuous lighting led to growth inhibition. Furthermore, the interaction between light intensity and photoperiod significantly affected plant height, root length, leaf area, and DQI in both cultivars. Root length, which plays a crucial role in plantlet growth and environmental adaptation, was greatest under 150 µmol m^−2^ s^−1^ with 20 h d^−1^, reaching 13.4 cm and 15.6 cm for Charlotte and Auto Charlotte, respectively. Both lower (50 µmol m^−2^ s^−1^) and higher (200 µmol m^−2^ s^−1^) light intensities suppressed root elongation. Regarding leaf number and leaf area, both cultivars performed best under the photoperiod of 20 h d^−1^, which facilitates photon capture and enhances photosynthetic efficiency. The highest leaf number (10.7) and leaf area (57.5 cm^2^) for Charlotte were observed at 100 µmol m^−2^ s^−1^, while Auto Charlotte exhibited the highest leaf number (10.7) and leaf area (25.1 cm^2^) at 150 µmol m^−2^ s^−1^. Compared with L200-P24, the leaf number increased by 18.9% and 39.0%, and the leaf area increased by 168.7% and 93.1% for Charlotte and Auto Charlotte, respectively. Under light conditions of 100–150 µmol m^−2^ s^−1^ with 20 h d^−1^, Charlotte exhibited a higher DQI. Auto Charlotte showed a similar increase under the same conditions, although its response to photoperiod changes was less pronounced, likely due to its day-neutral nature.

### 3.3. Biomass

In the Charlotte cultivar, the photoperiod had a greater effect on biomass than light intensity (Figure 2). Under the photoperiod of 20 h d^−1^, the root and shoot biomass of the plantlets were relatively higher. Specifically, at light intensity of 100 µmol m^−2^ s^−1^, the maximum values were observed, with root and shoot fresh weights of 0.68 g and 1.58 g, and dry weights of 0.04 g and 0.20 g, respectively. These values were significantly higher than those in other treatments. In contrast, in the Auto Charlotte cultivar, light intensity had a greater impact on biomass than photoperiod (Figure 2). Biomass showed an initial increase followed by a decline as light intensity increased. The maximum root and shoot biomass were observed under a light intensity of 100–150 µmol m^−2^ s^−1^. Additionally, the root and shoot biomass of Charlotte exceeded that of Auto Charlotte, likely because of the day-neutral nature of Auto Charlotte, which tends to trigger earlier reproductive development, thereby reallocating assimilates away from vegetative biomass accumulation.

### 3.4. Photosynthetic Pigment Content

The influence of photoperiod on the photosynthetic pigment content of plantlets was more pronounced than that of light intensity (Figure 3). In the Charlotte cultivar, chlorophyll a and chlorophyll b contents under the L150-P16 treatment reached 1.39 and 0.53 mg g^−1^, respectively. These values were significantly higher than those under other treatments, though not significantly different from the L150-P20 treatment. The highest carotenoid content (0.25 mg g^−1^) was observed under L150-P20, whereas continuous lighting treatments showed markedly reduced carotenoid levels, with a maximum decline of 68%. In the Auto Charlotte cultivar, the L050-P16 treatment resulted in the highest chlorophyll a and b contents at 2.05 and 0.96 mg g^−1^, respectively. Conversely, the lowest values were recorded under the L200-P24 treatment, with significant reductions of 65.4% and 71.9%, respectively. The carotenoid content followed a similar trend to that of chlorophylls, with relatively higher levels under shorter photoperiods compared to continuous lighting.

### 3.5. Antioxidant Enzyme Activity

For both cultivars, SOD activity declined with increasing light intensity under a photoperiod of 16 h d^−1^, indicating lower reactive oxygen species (ROS) levels under this photoperiod condition and consequently reduced demand for SOD activation (Figure 4). In contrast, under the photoperiod of 18 and 24 h d^−1^, SOD activity exhibited an increasing trend with rising light intensity. Under the conditions of 200 µmol m^−2^ s^−1^ and 24 h d^−1^, the SOD activity of Charlotte and Auto Charlotte plantlets reached maximum values of 817 and 780 U g^−1^ FW^−1^, respectively, indicating a severe oxidative stress response triggered by excessive light exposure. Therefore, plantlets under continuous lighting conditions may experience significant light stress and respond by activating SOD to counteract superoxide anions. In addition, the POD activity of both cultivars was highest under 20 h d^−1^, but it showed a decreasing trend as light intensity increased. Meanwhile, CAT activity was highest under 16 h d^−1^, significantly exceeding the levels observed under continuous lighting. This may be attributed to excessive ROS generation induced by prolonged exposure to lighting, leading to an upregulation of SOD activity to convert superoxide anions into hydrogen peroxide. The subsequent accumulation of hydrogen peroxide may alter ROS detoxification pathways, thereby lowering POD and CAT activities.

### 3.6. Root Activity

Moderate increases in light intensity combined with extended photoperiods significantly enhanced root activity in both cultivars (Figure 5). Under a constant photoperiod, root activity increased with rising light intensity; however, in the Charlotte cultivar, this promotive effect was suppressed when the intensity reached 200 µmol m^−2^ s^−1^. Conversely, when light intensity was held constant, extending the photoperiod from 16 to 20 h d^−1^ significantly improved root activity. Specifically, the Charlotte cultivar exhibited the highest root activity at 150 µmol m^−2^ s^−1^, reaching 88.2 μg TTF g^−1^ h^−1^, while Auto Charlotte peaked at 200 µmol m^−2^ s^−1^, achieving 271.2 μg TTF g^−1^ h^−1^.

### 3.7. Net CO_2_ Exchange Amount

In this study, the effect of light intensity on net CO_2_ exchange rate (NCER) was evaluated under the photoperiod of 20 h d^−1^ (Figure 6A,B). The results showed that Charlotte exhibited the highest Light NCER at 100 µmol m^−2^ s^−1^, averaging 0.08 µmol min^−1^, whereas Auto Charlotte peaked at 150 µmol m^−2^ s^−1^, averaging 0.15 µmol min^−1^. However, when light intensity increased to 200 µmol m^−2^ s^−1^, Light NCER decreased by 85.4% in Charlotte and 57.7% in Auto Charlotte. Additionally, the Dark NCER values for the two cultivars were lowest under 50 µmol m^−2^ s^−1^, measured at 0.05 and 0.26 µmol min^−1^, respectively. When light intensity increased to 200 µmol m^−2^ s^−1^, these values increased by 180% and 73.1%, respectively. Under the photoperiod of 20 h d^−1^, increasing light intensity initially enhanced the light NCER of both cultivars, followed by a decline at higher intensities. In contrast, dark NCER showed a continuous upward trend with increasing light intensity (Figure 6C,D). Notably, as light intensity increased from 50 to 100–150 µmol m^−2^ s^−1^, the Light NCEA of Charlotte and Auto Charlotte increased by 41.5% and 204.9%, respectively.

### 3.8. Cluster Heatmap Analysis

The cluster heatmap clearly illustrates the impact of different combinations of light intensity and photoperiod on the growth parameters, biomass, photosynthetic pigment content, root activity, and antioxidant enzyme activities of the plantlets. Cluster analysis was performed on the dataset, revealing two primary clusters for both cultivars (Figure 7). For Charlotte, L050-P20, L100-P16, L100-P20, L150-P16, L200-P20, and L150-P20 were grouped into one main cluster. This cluster was characterized by greater plant height, longer roots, larger biomass, higher photosynthetic pigment content, and enhanced root activity. In Auto Charlotte, L150-P20, L100-P20, L200-P20, and L100-P16 were grouped into one primary cluster, exhibiting similar traits, such as greater plant height, longer roots, larger biomass, and stronger root activity. However, under conditions of continuous lighting and high light intensity, both cultivars showed opposite trends in growth performance.

## 4. Discussion

### 4.1. Moderate Increases in Light Intensity and Photoperiod Promote Growth and Rooting

Cannabis exhibits distinct light requirements at different stages. In this study, we investigated the effects of four light intensities and three photoperiods on the growth and rooting of two medicinal cannabis cultivars in PAM. The results demonstrated that moderate increases in light intensity and photoperiod significantly enhanced biomass accumulation in both cultivars. Specifically, Charlotte showed the highest biomass under the L100-P20 treatment, while Auto Charlotte achieved maximum biomass under L150-P20. These results are consistent with those of Zarei et al. [2], who reported improved rooting and biomass accumulation in cannabis cuttings exposed to 150 μmol m^−2^ s^−1^ light intensity and an 18 h d^−1^ photoperiod. Moreover, as shown in Table 2, both cultivars exhibited a similar pattern, where plant height, root length, leaf number, leaf area, and DQI initially increased and then declined with increasing light intensity under the same photoperiod. These morphological and quality parameters peaked at 100 or 150 μmol m^−2^ s^−1^. This trend aligns with the findings of Murphy and Adelberg [29], who observed enhanced leaf development and improved root systems in cannabis plantlets with increasing light intensity from 25 to 167 μmol m^−2^ s^−1^ under a 14 h d^−1^ photoperiod. Taken together, our findings suggest that a light intensity of 100–150 μmol m^−2^ s^−1^ combined with a 20 h d^−1^ photoperiod is optimal for PAM of medicinal cannabis, as it facilitates robust vegetative development, increases leaf number and area, and significantly enhances biomass accumulation in plantlets.

### 4.2. Continuous Lighting Is Not Conducive to Cannabis Plantlets in PAM

To maintain cannabis in the vegetative growth stage for extended periods (from cloning to transplanting), long photoperiods are typically employed (generally exceeding 12 h, with 16–18 h being ideal, and even up to 24 h in some cases) [26,30]. Some genotypes may benefit from longer photoperiods, which enhance photosynthesis and thereby promote vegetative growth, including increased plant height, stem elongation, and dry biomass accumulation [45]. In this study, we analyzed the effects of three different photoperiods (16, 20, and 24 h d^−1^) on the growth and rooting of medicinal cannabis in PAM. The results indicated that plant height, root length, leaf number, and leaf area were highest under 16 and 20 h d^−1^ photoperiods, while the continuous lighting resulted in the poorest overall performance (Figure 6). As shown in Figure 2, Charlotte plantlets achieved the highest biomass under the 20 h d^−1^ photoperiod, while continuous lighting resulted in the lowest biomass. For Auto Charlotte, fresh weight peaked under the 20 h d^−1^ photoperiod, whereas the dry weight was highest under 24 h d^−1^. This difference is likely attributable to its day-neutral nature, making it less sensitive to photoperiod changes and more capable of converting prolonged lighting into dry matter accumulation.

Numerous studies have investigated the effects of continuous lighting on plant growth and development, with results varying widely across species [46,47]. While some plants exhibit increased biomass under continuous lighting, it can also lead to several detrimental effects, such as leaf chlorosis and induced light stress responses [48]. Our study demonstrated that continuous lighting is unfavorable for the growth of plantlets, as it significantly reduced chlorophyll a and b contents and caused visible leaf chlorosis (Figure 1). This aligns with the findings of Martina et al. [48], which showed that a photoperiod of 16 h d^−1^ was most favorable for cannabis growth. Continuous lighting, however, caused damage to thylakoid membranes, thereby causing a decrease in chlorophyll and carotenoid contents. Previous studies have further demonstrated that continuous lighting induces electron donor deficiency and promotes ROS accumulation, thereby disrupting photosynthesis and ultimately leading to oxidative damage [46,49]. Thylakoid membrane-associated compounds, such as carotenoids, serve as the first line of defense against ROS [50], while increased antioxidant enzyme activity rapidly scavenges these harmful substances [51]. In our study, plantlets subjected to continuous lighting exhibited the highest SOD activity, while POD and CAT activities were lowest. This indicates that continuous lighting induced a light stress response, prompting ROS production and an elevation in SOD activity. In turn, SOD converted superoxide anions into hydrogen peroxide, subsequently decreasing POD and CAT activities. Moreover, root activity, an important indicator of plantlet growth and health, was enhanced by extending the photoperiod in both cultivars, with the highest observed under the 20 h d^−1^ photoperiod. However, under continuous lighting, this promotive effect was diminished. Based on these findings, we recommend using a 20 h d^−1^ photoperiod for PAM of medicinal cannabis, as it promotes better growth and rooting, while continuous lighting should be avoided.

### 4.3. The Influence of Light Intensity on Net CO_2_ Exchange Amount

NCER is a crucial physiological indicator reflecting the plant’s ability to sequester carbon and release oxygen, providing key insights into plant biomass accumulation, carbon balance dynamics, and environmental adaptability [50,52,53]. As early as 1998, Niu et al. [43] designed a system to measure the CO_2_ exchange rate (net photosynthetic rate and dark respiration rate) of plantlets in naturally ventilated containers. This method has since been widely applied in numerous studies, including cannabis [54,55] and wasabi plantlets [56,57]. In this study, we analyzed the effect of light intensity on the NCER in PAM of medicinal cannabis under a 20 h d^−1^ photoperiod. As shown in Figure 6C Charlotte plantlets had the highest light NCER at 100 μmol m^−2^ s^−1^, while the lowest was observed at 200 μmol m^−2^ s^−1^. Auto Charlotte plantlets exhibited a higher net exchange rate at 150 μmol m^−2^ s^−1^, but this decreased as light intensity increased to 200 μmol m^−2^ s^−1^. This may be due to low CO_2_ concentration in the GA-7, which prevents further enhancement of net photosynthesis of plantlets at higher light intensities [58]. These results align with those of Marco et al. [54], who reported that the light saturation point of cannabis explants was lower when CO_2_ concentration was limited. However, when the CO_2_ concentration was 1200 μmol mol^−1^, the NCER continued to rise with increasing light intensity up to 400 μmol m^−2^ s^−1^ without reaching the light saturation point. PAM typically requires a higher air exchange rate to supply the necessary CO_2_ for photosynthesis [55]. Additionally, studies have revealed that elevating CO_2_ concentration is particularly critical for promoting plant growth and development during micropropagation. This is because increased CO_2_ significantly enhances photosynthesis in cannabis plantlets, thereby boosting leaf number, biomass, and meristematic activity [24,59]. Our results also show that increasing light intensity enhanced the light NCEA of both cultivars, thereby simultaneously improving dry matter accumulation. These results align with previous studies, which reported that increased daily carbon gain contributed to a 29–31% increase in biomass, while insufficient CO_2_ supply during the light period inhibited photosynthetic activity [60,61]. Notably, some studies have shown that optimizing the air exchange rate of culture vessels and enhancing CO_2_ concentration in the culture room can effectively improve the photosynthetic capacity of explants [29,55,62]. Therefore, future research could focus on further optimizing these factors to enhance the growth potential of cannabis plantlets in PAM.

## 5. Conclusions

This study optimized light intensity and photoperiod in the PAM of medicinal cannabis to enhance carbon assimilation and promote plantlet growth, thereby improving seedling quality. The results demonstrated that under an LED lighting condition with a light intensity of 100–150 µmol m^−2^ s^−1^ and a photoperiod of 20 h d^−1^, both cultivars exhibited maximum leaf number, leaf area, root length, biomass, and root activity, whereas continuous lighting had an inhibitory effect on these parameters. Therefore, this optimized lighting condition is recommended for the PAM of medicinal cannabis. In future research, elevating CO_2_ concentration and increasing air exchange rate in culture vessels are expected to further enhance plantlets’ photosynthetic carbon assimilation, thereby promoting robust growth and improving overall seedling quality.

## Figures and Tables

**Figure 1 biology-14-00706-f001:**
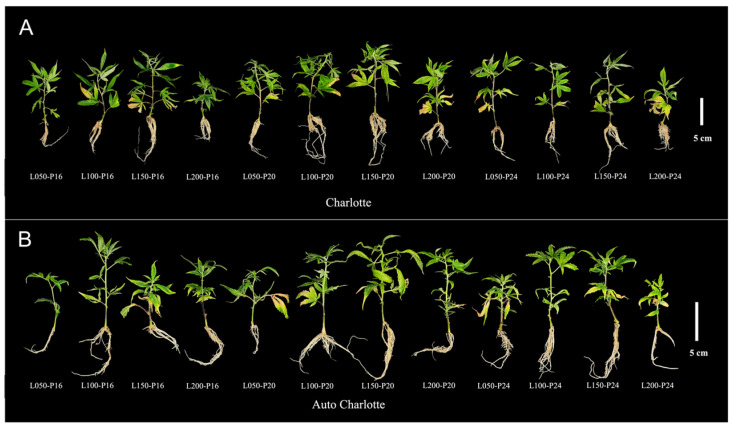
Growth status of Charlotte (**A**) and Auto Charlotte (**B**) plantlets under different combinations of light intensity and photoperiod. L represents light intensity, and P represents photoperiod.

**Figure 2 biology-14-00706-f002:**
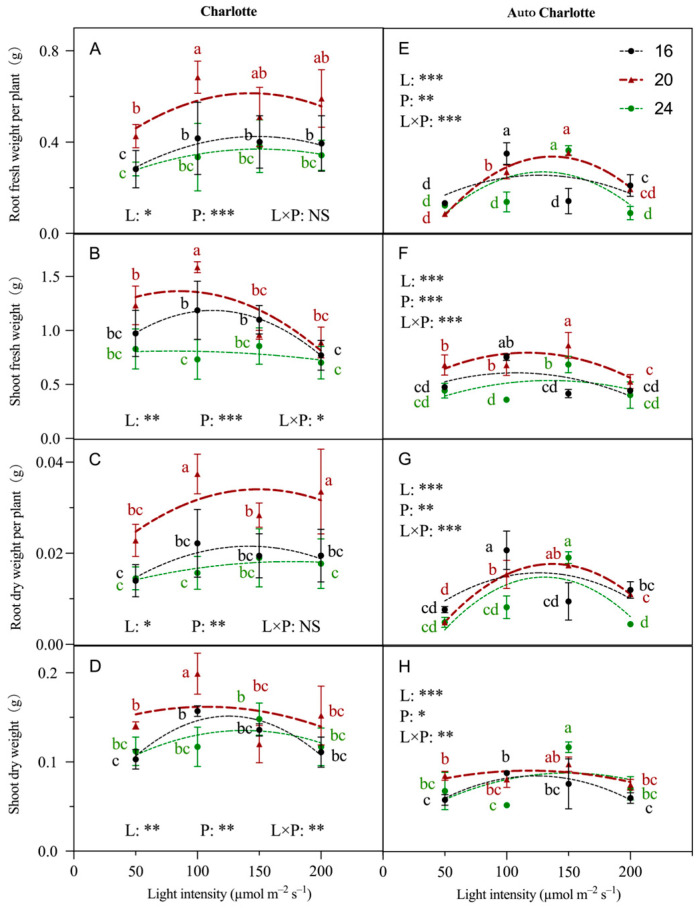
Effects of light intensity and photoperiod on the root fresh weight, shoot fresh weight, root dry weight and shoot dry weight of plantlets of Charlotte (**A**–**D**) and Auto Charlotte (**E**–**H**). The numbers 16, 20, and 24 represent photoperiod levels (h d^−1^). Identical letters indicate no significant difference between group means, while different letters indicate significant differences. L represents light intensity, P represents photoperiod, and L × P denotes the interaction between light intensity and photoperiod. Statistical significance levels are indicated by * for *p* < 0.05, ** for *p* < 0.01, and *** for *p* < 0.001, while NS represents no significant effect.

**Figure 3 biology-14-00706-f003:**
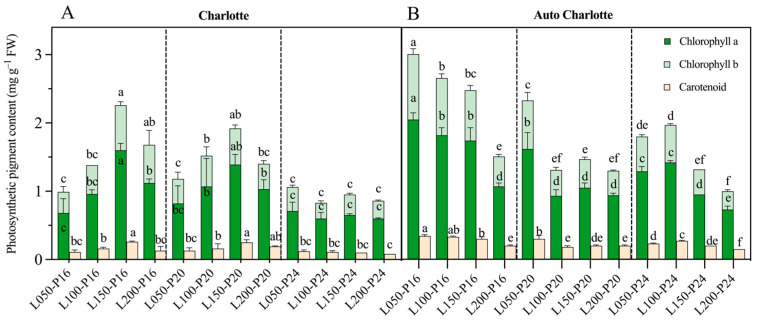
Effects of light intensity and photoperiod on the photosynthetic pigment content of plantlets of Charlotte (**A**) and Auto Charlotte (**B**). L represents light intensity, and P represents photoperiod. The light intensity levels are denoted by values of 50, 100, 150, and 200 µmol m^−2^ s^−1^, while the photoperiod levels are represented by 16, 20, and 24 h d^−1^. Identical letters indicate no significant difference between group means, while different letters denote significant differences.

**Figure 4 biology-14-00706-f004:**
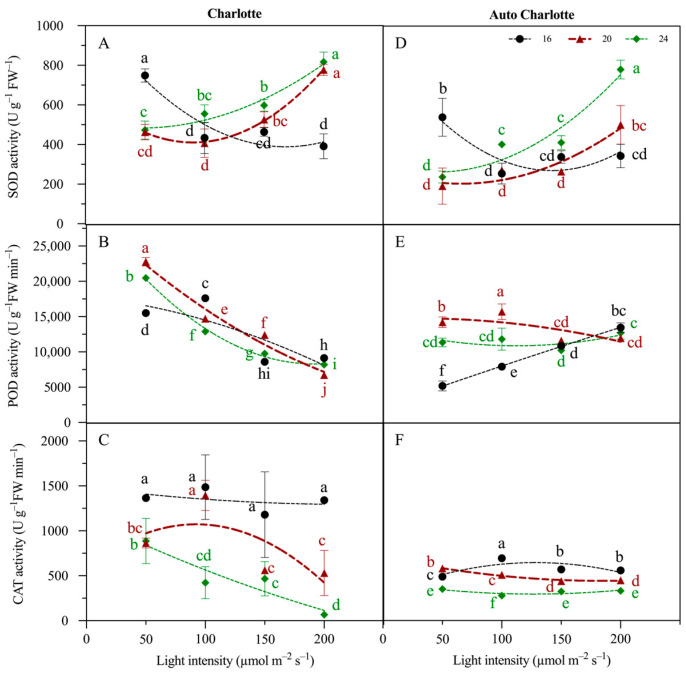
Effects of light intensity and photoperiod on superoxide dismutase (SOD), peroxidase (POD) and catalase (CAT) activity of plantlets of Charlotte (**A**–**C**) and Auto Charlotte (**D**–**F**). Identical letters indicate no significant difference between group means, while different letters denote significant differences.

**Figure 5 biology-14-00706-f005:**
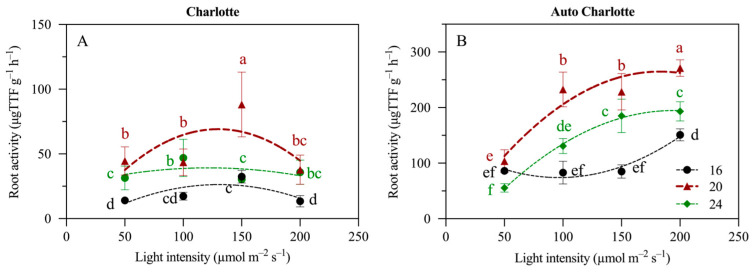
Effects of light intensity and photoperiod on the root activity of the plantlets of Charlotte (**A**) and Auto Charlotte (**B**). Identical letters indicate no significant difference between group means, while different letters denote significant differences.

**Figure 6 biology-14-00706-f006:**
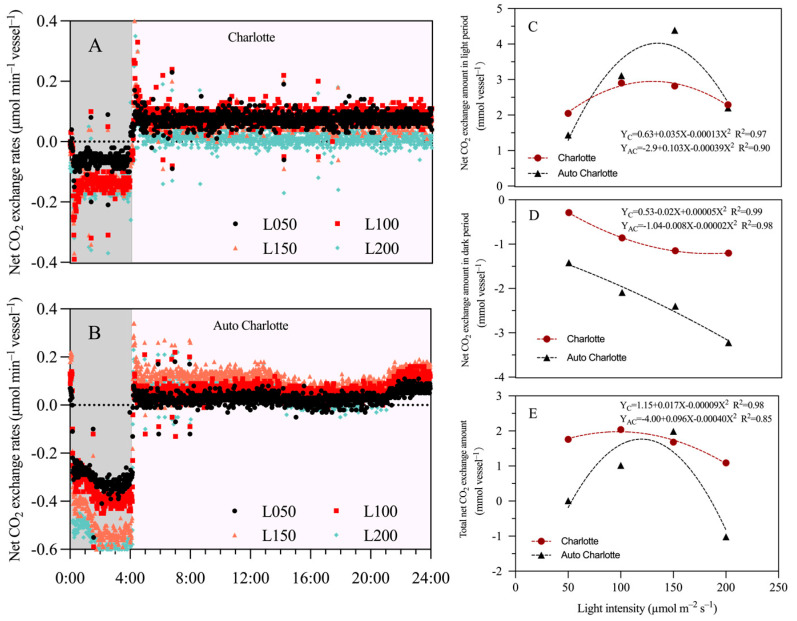
Effects of light intensity on the net CO_2_ exchange rate (**A**,**B**) at 14 d after inoculation and the total net CO_2_ exchange amount (**C**–**E**) over the entire culture period for plantlets of two cultivars under the photoperiod of 20 h d^−1^. L represents light intensity, with 50, 100, 150, and 200 indicating levels of µmol m^−2^ s^−1^. Gray represents the dark period, while pink denotes the light period.

**Figure 7 biology-14-00706-f007:**
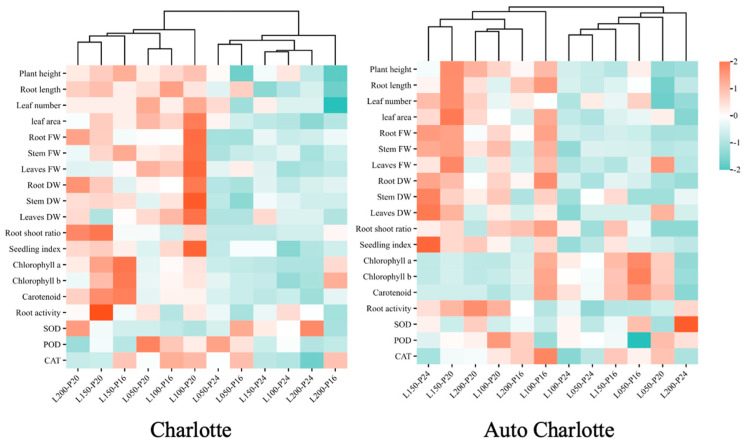
Clustering heatmap of all measured parameters for the two cultivars. L represents light intensity, with 50, 100, 150, and 200 indicating levels of µmol m^−2^ s^−1^, and P represents photoperiod, with 16, 20, and 24 indicating levels of h d^−1^. Red denotes parameter values above the dataset average, while green denotes values below the average, with color intensity reflecting the magnitude of deviation. FW and DW represent fresh weight and dry weight, respectively.

**Table 1 biology-14-00706-t001:** Light environment conditions for each treatment.

Treatment	Light Intensity (μmol m^−2^ s^−1^)	Actual Light Intensity (μmol m^−2^ s^−1^)			Photoperiod (h d^−1^)	Daily Light Integral (mol m^−2^ d^−1^)
L050-P16	50	53.5	±	0.8	16	3.1
L100-P16	100	105.0	±	0.3	16	6.0
L150-P16	150	152.8	±	1.2	16	8.8
L200-P16	200	190.5	±	5.9	16	11.0
L050-P20	50	52.1	±	1.9	20	3.8
L100-P20	100	102.4	±	1.9	20	7.4
L150-P20	150	151.8	±	3.3	20	10.9
L200-P20	200	196.4	±	7.5	20	14.1
L050-P24	50	53.7	±	1.4	24	4.6
L100-P24	100	103.2	±	2.3	24	8.9
L150-P24	150	152.8	±	1.2	24	13.2
L200-P24	200	193.2	±	2.8	24	16.7

**Table 2 biology-14-00706-t002:** Growth parameters of plantlets. L represents light intensity, P represents photoperiod, and L × P denotes the interaction between light intensity and photoperiod. The light intensity levels are denoted by values of 50, 100, 150, and 200 µmol m^−2^ s^−1^, while the photoperiod levels are represented by 16, 20, and 24 h d^−1^. Identical letters indicate no significant difference between group means, while different letters denote significant differences. Statistical significance levels are indicated by * for *p* < 0.05, ** for *p* < 0.01, and *** for *p* < 0.001, while NS represents no significant effect.

Cultivar	Treatment	Plant Height (cm)	Root Length (cm)	Stem Diameter (mm)	Number of Leaves	Leaf Area (cm^2^)	Water Content (%)	Root Shoot Ratio	Seedling Index (DQI)
Charlotte	L050-P16	6.9 ± 1.7 c	12.7 ± 2.3 ab	2.2 ± 0.3 ab	8.3 ± 0.6 bc	30.0 ± 5.7 c	90.5 ± 1.3 a	0.13 ± 0.02 c	0.0032 ± 0.0009 b
	L100-P16	10.7 ± 1.2 ab	14.8 ± 0.7 a	2.3 ± 0.1 ab	9.7 ± 0.6 ab	43.6 ± 13.2 b	88.4 ± 2.3 b	0.14 ± 0.04 c	0.0036 ± 0.0004 ab
	L150-P16	11.8 ± 1.3 a	11.3 ± 0.6 bc	2.2 ± 0.2 ab	9.7 ± 1.5 ab	39.1 ± 2.6 bc	89.6 ± 0.6 ab	0.16 ± 0.01 bc	0.0034 ± 0.0011 b
	L200-P16	10.2 ± 1.1 ab	5.4 ± 1.0 e	2.3 ± 0.1 ab	7.3 ± 0.6 c	26.7 ± 3.6 c	88.7 ± 0.8 b	0.17 ± 0.04 bc	0.0029 ± 0.0003 b
	L050-P20	6.6 ± 1.3 c	12.0 ± 2.7 b	2.3 ± 0.2 ab	10.7 ± 1.5 a	48.2 ± 7.7 ab	90.1 ± 0.9 ab	0.16 ± 0.02 bc	0.0030 ± 0.0004 b
	L100-P20	11.3 ± 1.5 ab	11.7 ± 1.0 b	2.4 ± 0.3 a	10.7 ± 1.5 a	57.5 ± 2.1 a	89.6 ± 1.0 ab	0.19 ± 0.01 b	0.0046 ± 0.0006 a
	L150-P20	11.0 ± 0.8 ab	13.4 ± 0.8 ab	2.3 ± 0.1 ab	9.7 ± 0.6 ab	44.7 ± 10.8 b	90.0 ± 0.8 ab	0.25 ± 0.00 a	0.0037 ± 0.0011 ab
	L200-P20	10.3 ± 0.4 ab	12.7 ± 0.1 ab	2.2 ± 0.2 ab	9.7 ± 1.2 ab	35.8 ± 6.1 bc	87.5 ± 0.7 bc	0.24 ± 0.02 a	0.0036 ± 0.0010 ab
	L050-P24	9.9 ± 0.5 ab	9.3 ± 1.2 c	2.1 ± 0.2 b	10.0 ± 1.0 ab	38.3 ± 5.8 bc	88.5 ± 1.2 b	0.13 ± 0.02 c	0.0028 ± 0.0002 b
	L100-P24	10.4 ± 1.7 ab	7.7 ± 1.3 de	1.8 ± 0.3 c	9.0 ± 1.0 b	26.4 ± 3.8 c	87.2 ± 2.5 bc	0.14 ± 0.03 c	0.0024 ± 0.0001 b
	L150-P24	9.5 ± 1.4 ab	6.3 ± 0.3 e	2.2 ± 0.1 ab	9.7 ± 1.2 ab	27.4 ± 2.0 c	86.3 ± 1.3 c	0.13 ± 0.03 c	0.0032 ± 0.0003 b
	L200-P24	8.4 ± 2.5 b	8.9 ± 1.1 d	2.0 ± 0.2 bc	9.0 ± 1.0 b	21.4 ± 5.1 d	87.1 ± 0.5 c	0.14 ± 0.01 c	0.0027 ± 0.0003 b
	L	**	**	NS	NS	**	*	**	NS
	P	*	***	**	*	***	**	***	***
	L × P	*	***	NS	NS	*	NS	NS	***
Auto Charlotte	L050-P16	6.6 ± 0.3 d	8.2 ± 1.7 c	2.0 ± 0.1 a	9.7 ± 0.6 ab	17.5 ± 1.2 c	89.2 ± 0.9 ab	0.13 ± 0.02 bc	0.0016 ± 0.0002 c
	L100-P16	9.1 ± 0.6 ab	12.6 ± 1.9 b	2.0 ± 0.2 ab	9.0 ± 1.0 ab	24.6 ± 3.1 ab	90.2 ± 0.4 ab	0.23 ± 0.05 a	0.0021 ± 0.0001 b
	L150-P16	6.1 ± 0.3 de	5.6 ± 0.5 de	1.9 ± 0.3 b	8.7 ± 1.2 ab	14.3 ± 0.9 cd	86.8 ± 2.3 b	0.20 ± 0.08 ab	0.0020 ± 0.0005 bc
	L200-P16	7.7 ± 0.1 bc	8.0 ± 1.7 c	1.8 ± 0.1 b	9.3 ± 0.6 ab	14.8 ± 1.0 c	89.1 ± 0.8 ab	0.20 ± 0.01 ab	0.0015 ± 0.0002 c
	L050-P20	5.5 ± 0.9 e	4.3 ± 0.6 e	1.7 ± 0.2 bc	7.3 ± 0.6 c	20.3 ± 1.9 b	88.2 ± 0.9 b	0.06 ± 0.01 d	0.0015 ± 0.0001 c
	L100-P20	8.5 ± 0.6 b	8.0 ± 0.5 c	1.8 ± 0.2 b	8.7 ± 1.5 ab	17.5 ± 0.9 c	89.8 ± 0.7 ab	0.19 ± 0.03 ab	0.0019 ± 0.0002 bc
	L150-P20	9.5 ± 0.3 ab	15.6 ± 1.4 a	1.9 ± 0.1 b	10.7 ± 0.6 a	25.1 ± 5.8 a	90.5 ± 0.3 a	0.18 ± 0.01 ab	0.0021 ± 0.0002 bc
	L200-P20	9.8 ± 0.4 a	10.7 ± 2.3 b	1.9 ± 0.2 b	9.7 ± 2.1 ab	19.5 ± 3.8 bc	88.1 ± 0.1 b	0.09 ± 0.00 c	0.0022 ± 0.0003 b
	L050-P24	6.4 ± 0.7 de	8.5 ± 0.4 c	1.8 ± 0.2 b	9.3 ± 1.2 ab	16.0 ± 2.8 c	87.7 ± 2.1 b	0.07 ± 0.01 d	0.0014 ± 0.0003 c
	L100-P24	7.2 ± 1.4 cd	6.3 ± 1.1 d	1.6 ± 0.1 c	8.0 ± 1.0 b	16.4 ± 4.3 c	87.9 ± 0.5 b	0.16 ± 0.04 b	0.0012 ± 0.0002 c
	L150-P24	7.3 ± 0.4 c	11.5 ± 0.2 b	1.8 ± 0.2 b	10.0 ± 1.0 ab	15.8 ± 3.7 c	87.1 ± 0.3 b	0.16 ± 0.02 b	0.0029 ± 0.0000 a
	L200-P24	6.3 ± 1.2 de	6.8 ± 0.6 ^cd^	1.6 ± 0.2 c	7.7 ± 0.6 bc	13.0 ± 1.0 d	84.7 ± 0.3 c	0.06 ± 0.01 d	0.0014 ± 0.0001 c
	L	***	***	NS	NS	NS	*	***	***
	P	***	*	*	NS	**	**	**	NS
	L × P	***	***	NS	*	**	*	NS	***

## Data Availability

Data will be made available on request.

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
