# Peer review of "Optimizing LED Light Intensity and Photoperiod to Promote Growth and Rooting of Medicinal Cannabis in Photoautotrophic Micropropagation"

_biology, 2025, doi:10.3390/biology14060706_

Round 1
Reviewer 1 Report
Comments and Suggestions for Authors
The article “Optimizing LED light intensity and photoperiod to promote growth and rooting of medicinal cannabis in photoautotrophic micropropagation” concerns an important species from an economic and medical point of view. Many questions have arisen while reading. I have some doubts about the experimental design and some aspects are not clear to me.
In my opinion, the manuscript cannot be accepted for publication in its current form and needs a major revision. Detailed comments can be found below.
Abstract
l. 42: the abbreviation L200-P24 should be clarified when it appears for the first time.
In my opinion abstract is too detailed, it should be more concise and contain only the most important results.
Keywords:
l. 53: “hotoautotrophic” - ???
Materials and methods
l. 138-139: „After cleaning, the seeds were soaked for 24 h, then placed in petri dishes to promote germination”. In my opinion, it is not clear how the seeds were cleaned: with tap water? How were they soaked? In the water? Were they placed on sterile filter paper in Petri dishes?
l. 141-146: “The nutrient solution (…)” Was the composition of the solution based on the literature or on the author’s own idea? Is the solution commercially available?
l. 148-151: “The system comprised two ABS cultivation beds (…)” In my opinion, a photo should be included in the manuscript to make it clear to the readers what the cultivation system looks like.
l. 170: Why were the shoot tips collected only from female plants?
l. 178-181: I have doubts whether the culture in the shelves can provide stable conditions during the experiment. Normally we conduct such experiments in a phytotron chamber where light intensity and photoperiod are controlled.
l. 223-229: the method for determining the content of photosynthetic pigments is quite original and is not often used. I have doubts whether the method is correct. What about the temperature? It is generally known that photosynthetic pigments are susceptible to temperature. The literature cited is not easily accessible (this comment concerns positions 35, 37 and 39 in References).
l. 248-251: I wonder on the aseptic condition inside the culture vessel? Is it possible to maintain sterility when you placed sensors inside the vessel?
I highly recommend to include the scheme of the experiment to clarify all the steps. There are many steps and combinations and the reader might be confused.
Results
Fig. 1. The signatures on the figure and the scale bars are illegible!
Fig. 2. Each individual graph on the figure should be labelled A, B, C, etc…
l. 361-362: “The highest carotenoid 361 content (0.25 mg g-1) was observed under L100-P20” – rather under L150-P20 (Fig. 3)
l. 366: it is difficult to check the % content when Fig. 3 says mg/g
Fig. 3 – please clarify mg/g FW or DW in the captions
Fig. 4 is not cited in the text, it should be self-explanatory, so I recommend describing the abbreviations for SOD, POD and CAT in the captions.
l. 379-381, l. 409-411 should be moved to the Discussion section
Author Response
- 1. L42: the abbreviation L200-P24 should be clarified when it appears for the first time.In my opinion abstract is too detailed, it should be more concise and contain only the most important results.
Response: Has been modified. The abstract has been streamlined to retain only the most important results, with less essential results moved to the Results section.
Keywords:
- 2. L53: “hotoautotrophic” - ???
Response: A missing letter "p" has been added, and the term has been corrected to "photoautotrophic".
- 3. L138-139: After cleaning, the seeds were soaked for 24 h, then placed in petri dishes to promote germination”. In my opinion, it is not clear how the seeds were cleaned: with tap water? How were they soaked? In the water? Were they placed on sterile filter paper in Petri dishes?
Response: The original text at L146-148 has been revised as follows:
"After cleaning with tap water, the seeds were soaked for 24 h, then placed in petri dishes containing sterile filter paper to promote germination. Sterile pure water was sprayed twice daily to keep the filter paper moist."
- L141-146: “The nutrient solution (…)” Was the composition of the solution based on the literature or on the author’s own idea? Is the solution commercially available?
Response: The N, P, and K concentrations in the nutrient solution were based on the research findings of Nirit Bernstein’s team, who conducted extensive experiments on mineral elements during the vegetative growth stage of Cannabis sativa. Their studies provided a foundation for optimizing nutrient formulations. After further calculation and adaptation, the nutrient solution was composed of the following compounds. Additionally, a citation to their research has been added to the manuscript to acknowledge the nutrient formulation foundation (L168-L169).
[35] Saloner A, Bernstein N. Dynamics of mineral uptake and plant function during development of drug-type medical cannabis plants [J]. Agronomy, 2023, 13(12): 2865. https://doi.org/10.3390/agronomy13122865
- L148-151: “The system comprised two ABS cultivation beds (…)” In my opinion, a photo should be included in the manuscript to make it clear to the readers what the cultivation system looks like.
Response: Photos of the mother plant cultivation system have been added to the supplementary materials to visually illustrate the described setup (Figure S1).
- L170: Why were the shoot tips collected only from female plants?
Response: In medical and pharmaceutical applications, only the flowers and leaves of female plants can be used to extract high-quality cannabinoids. Therefore, it is essential to ensure that all propagated plants are female. By selecting shoot tips (explants) only from female plants for propagation, this ensures that the culturing plantlets will also be female, which helps maintain the quality and consistency of cannabinoid production.
- L178-181: I have doubts whether the culture in the shelves can provide stable conditions during the experiment. Normally we conduct such experiments in a phytotron chamber where light intensity and photoperiod are controlled.
Response: Yes, the experiment was conducted in a controlled environment. We have clarified this in the revised manuscript at L204. All environmental parameters, including light intensity, temperature and relative humidity were precisely controlled throughout the experiment. The actual light intensity data are provided in Table 1 and demonstrate good uniformity across the culture shelves. Additionally, Figure S3 in the supplementary materials presents the real-time data for temperature, relative humidity, and CO2 levels, further supporting the stability of the growth conditions.
- L223-229: the method for determining the content of photosynthetic pigments is quite original and is not often used. I have doubts whether the method is correct. What about the temperature? It is generally known that photosynthetic pigments are susceptible to temperature. The literature cited is not easily accessible (this comment concerns positions 35, 37 and 39 in References).
Response: The method used for determining photosynthetic pigment content is based on well-established protocols from authoritative plant physiology experimental textbooks published by Chinese academic presses. These references are standard teaching materials widely used in Chinese universities and are readily accessible through platforms such as CNKI and university libraries using Chinese search terms (e.g., 李合生, 植物生理生化实验原理和技术 [M]. 高等教育出版社, 2000; 植物生理学实验指导). In response to the reviewer’s concern, we have revised References 35, 37, and 39 by italicizing the book titles to comply with Biology citation norms and improve clarity.
Regarding temperature sensitivity: if the concern is about pigment degradation during the extraction process, the protocol we followed specifies performing operations under low light conditions and carrying out pigment extraction in the dark to prevent chlorophyll degradation. The entire procedure was conducted at room temperature (20–22 °C), which is within the acceptable range that does not affect the stability of the pigments during the extraction process. The method itself does not indicate temperature as a variable significantly affecting the measurement results under these conditions.
- L248-251: I wonder on the aseptic condition inside the culture vessel? Is it possible to maintain sterility when you placed sensors inside the vessel?
Response: The culture vessels can maintain aseptic conditions through a rigorous sterilization protocol. Explants, vessels, and supporting materials are all subjected to sterilization prior to use.
During sensor installation, sensor surfaces are disinfected, and the procedure is performed under sterile conditions. Additionally, the culture room undergoes regular ultraviolet (UV) irradiation to ensure sterility. While occasional colony growth may occur in large-scale propagation, the sugar-free composition of the medium minimizes microbial proliferation, and any impact on plantlets growth remains negligible.
- I highly recommend to include the scheme of the experiment to clarify all the steps. There are many steps and combinations and the reader might be confused.
Response: Thank you for the thoughtful suggestion. We agree that visual aids such as schematic diagrams can be helpful in complex experimental designs. However, in this study, we have already described the experimental procedures in a sequential and structured manner, following the logical flow of operations:
- Mother plant cultivation – for explant procurement;
- Culture medium preparation – including vessel type, support materials, nutrient composition, and PGR concentrations;
- Explant acquisition and sterilization – with details on explant size, sterilization protocol, and aseptic inoculation;
4)Culture environment setup – covering controlled conditions for light, temperature, relative humidity, and CO2.
We believe this detailed step-by-step structure provides sufficient clarity for readers to follow the experimental process without ambiguity. Therefore, we respectfully consider the current text to be adequate for understanding the methodology.
Results
- 1. The signatures on the figure and the scale bars are illegible!
Response: Thank you for pointing this out. Figure 1 has been revised to improve overall clarity. The labels and scale bars have been enlarged and enhanced to ensure they are fully legible. Additionally, to streamline the presentation and improve readability, some less essential photos have been moved to the supplementary materials.
- 2. Each individual graph on the figure should be labelled A, B, C, etc…
Response: Figure 2 has been revised, and labels (A, B, C, etc.) have been added to each individual graph to clearly distinguish the subfigures.
- L361-362: “The highest carotenoid 361 content (0.25 mg g-1) was observed under L100-P20” – rather under L150-P20 (Fig. 3)
Response: Thank you for pointing out this inconsistency. The manuscript has been corrected accordingly. In the Charlotte cultivar, the highest carotenoid content (0.25 mg g-1) was observed under the L150-P20 treatment (L443-444).
- L366: it is difficult to check the % content when Fig. 3 says mg/g
Fig. 3 – please clarify mg/g FW or DW in the captions
Response: Has been modified. The units in Figure 3 have been clarified in the figure caption to specify whether the values are expressed as mg per gram of fresh weight (mg/g FW). We further calculated the percentage through the use of measurement results.
- 4 is not cited in the text, it should be self-explanatory, so I recommend describing the abbreviations for SOD, POD and CAT in the captions.
Response: Has been modified. The abbreviations for SOD (superoxide dismutase), POD (peroxidase), and CAT (catalase) have been added and clearly described in the caption of Figure 4 to ensure the figure is self-explanatory.
- L379-381, L409-411 should be moved to the Discussion section
Response: Has been modified. The content originally at Lines 369–372, 379–381, and 409–411 has been removed from the Results section and incorporated into the Discussion section accordingly.

Reviewer 2 Report
Comments and Suggestions for Authors
Dear Authors,
This study addresses an important topic within plant biotechnology — optimizing environmental conditions for the photoautotrophic micropropagation (PAM) of medicinal cannabis. The research investigates the effects of varying light intensity and photoperiod on two cultivars of Cannabis sativa L., namely 'Charlotte' and 'Auto Charlotte'. The manuscript is well-structured, with a clear introduction, detailed methodology, comprehensive results, and relevant discussion. It contributes valuable data for improving PAM protocols for cannabis, which is increasingly significant due to its medical and economic importance. It should be noted, that цith the rising demand for standardized, disease-free cannabis seedlings for medical use, this study fills a critical gap by proposing optimized light conditions under aseptic and sugar-free culture systems. The experimental design is robust, involving multiple light intensities (50–200 µmol m⁻² s⁻¹) and photoperiods (16–24 h d⁻¹), providing a thorough exploration of optimal conditions for cannabis propagation. The authors used cluster analysis and heatmap visualization to integrate various physiological, biochemical, and morphological parameters, offering a holistic view of plantlet performance. The conclusion that 100–150 µmol m⁻² s⁻¹ light intensity combined with a 20 h d⁻¹ photoperiod enhances root development, biomass accumulation, and photosynthetic efficiency provides actionable guidance for tissue culture practices in cannabis cultivation.
But there are some minor comments on the scientific research:
- The abstract is brief but could have better emphasized the novelty and practical implications of the study. It may be worth emphasizing how the proposed light conditions improve PAM over traditional methods and their potential scalability for commercial production.
- The introduction effectively contextualizes the problem, but there is no strong hypothesis or objective statement at the end. It is necessary to clearly indicate the purpose of the study in the last paragraph of the introduction, for example: «The purpose of this study was to determine the optimal combination of light intensity and photoperiod to improve rooting and growth in photoautotrophic micropropagation of medicinal cannabis».
- The methods are generally well-documented, but some details require clarification: сlarify whether the LED lamps provided consistent spectral composition across all treatments. Specify the calibration method and frequency for CO₂ sensors to ensure measurement accuracy. Mention if any corrections were applied for multiple comparisons (e.g., Bonferroni correction).
- The presentation of results is mostly clear, but there is room for improvement in visual clarity– some figures (e.g., Figure 1) lack sufficient detail in labels and resolution. Ensure high-resolution images and clearly labeled axes. Tables should be simplified to avoid redundancy. For example, Table 2 contains many statistical annotations; consider using footnotes or a separate table legend instead of cluttering the main body.
- The discussion is thorough but occasionally repetitive. Avoid reiterating results. Instead, focus on interpreting the findings in the context of previous studies and explaining the mechanisms behind observed responses (e.g., why continuous lighting causes chlorosis or oxidative stress). Expand on how these findings can be integrated into commercial-scale tissue culture systems , including possible limitations and scalability issues.
- The conclusion is brief and summarizes the key findings. Add a brief note about future research directions, such as studying other varieties, testing different concentrations of CO₂, or evaluating the success of acclimatization after micro-breeding.
Overall, the English is good, but some sentences are too long or difficult to compose. Tip: Subtract them for minor grammatical improvements and sentence structure optimization. For example: The original: «However, these methods rely on the use of sealed containers and sugar-containing media, which leads to high production costs, low rooting success rates, and low graft survival». Improved: «However, these methods rely on the use of sealed containers and sugar-containing media, resulting in high production costs, low rooting success rates, and poor graft survival».
Overall, with minor changes regarding the above issues, this manuscript will make a significant contribution to the field of plant biotechnology and cannabis research. The data and conclusions have practical applications for both academic and industrial purposes.
The manuscript as a whole is clear and clearly reflects the objectives of the research. However, it contains numerous grammatical errors, inconsistent formatting, and awkward wording that can make it difficult to read and reduce the quality of the work. A small professional revision in English is required before publication in an international journal.
After the revision, the article can be accepted for publication.
Comments on the Quality of English LanguageA small professional revision in English is required before publication in an international journal. With appropriate language editing and formatting improvements, this manuscript will be well-suited for publication in an international journal. Many of the issues identified are correctable through careful proofreading and revision.
Author Response
- The abstract is brief but could have better emphasized the novelty and practical implications of the study. It may be worth emphasizing how the proposed light conditions improve PAM over traditional methods and their potential scalability for commercial production.
Response: The abstract has been revised accordingly (Lines 57–61) to better emphasize the practical implications of the study. Specifically, it now highlights that these findings provide essential technical support for the large-scale propagation of vigorous, disease-free female plantlets with well-developed root systems and high genetic uniformity, thereby meeting the stringent quality standards for planting materials in the commercial cultivation of cannabis for medical and pharmaceutical use.
- The introduction effectively contextualizes the problem, but there is no strong hypothesis or objective statement at the end. It is necessary to clearly indicate the purpose of the study in the last paragraph of the introduction, for example: «The purpose of this study was to determine the optimal combination of light intensity and photoperiod to improve rooting and growth in photoautotrophic micropropagation of medicinal cannabis».
Response: We appreciate your feedback on the introduction. In the original manuscript, the study’s objective is stated in Lines 138–140 of the Introduction: "This study aims to explore the optimal combination of light intensity and photoperiod suitable for the photoautotrophic micropropagation of medicinal cannabis, with the goal of improving the overall quality of plantlets." We believe this statement effectively conveys the purpose of the study as you recommended.
- The methods are generally well-documented, but some details require clarification: сlarify whether the LED lamps provided consistent spectral composition across all treatments. Specify the calibration method and frequency for CO2sensors to ensure measurement accuracy. Mention if any corrections were applied for multiple comparisons (e.g., Bonferroni correction).
Response: Regarding the LED lamps, as mentioned in lines L212-213 of the original manuscript, all treatments used identical white LED lamps (W4000K - 18 W, with an R:B ratio of 1.8, provided by Beijing Lighting Valley Technology Co., Ltd., Beijing, China). These lamps were installed 15 cm above the GA-7 and served as the artificial lighting source. This setup ensured consistent spectral composition and light quality across all experimental treatments.
For the calibration of CO2 sensors, as supplemented in L304-308. (Before the experiment and at weekly intervals during the experiment, all sensors were placed in a standardized reference environment to record CO2 concentration, temperature, and humidity over a 2 h period. Average values for each parameter were calculated, and experimental data were corrected by comparing differences between these average values, ensuring data accuracy. ).
- The presentation of results is mostly clear, but there is room for improvement in visual clarity– some figures (e.g., Figure 1) lack sufficient detail in labels and resolution. Ensure high-resolution images and clearly labeled axes. Tables should be simplified to avoid redundancy. For example, Table 2 contains many statistical annotations; consider using footnotes or a separate table legend instead of cluttering the main body.
Response: Thank you for your helpful suggestions.
Figure 1 has been updated with higher-resolution images, and the scale bars and labels have been enhanced for better clarity. Additionally, some non-essential figures have been moved to the supplementary materials.
Table 2 has been reformatted for improved readability—statistical annotations are now presented as superscript letters to reduce clutter in the table body.
- The discussion is thorough but occasionally repetitive. Avoid reiterating results. Instead, focus on interpreting the findings in the context of previous studies and explaining the mechanisms behind observed responses (e.g., why continuous lighting causes chlorosis or oxidative stress). Expand on how these findings can be integrated into commercial-scale tissue culture systems , including possible limitations and scalability issues.
Response: Thank you for your insightful comments. We have integrated the sections in the discussion on leaf chlorosis caused by continuous light and cited previous literature (L608-616) to explain the causes of leaf chlorosis. Additionally, I revised L616-617 to avoid repetition with the results section. Moreover, we carefully reviewed the entire Results and Discussion sections to ensure that findings are not redundantly presented.
- The conclusion is brief and summarizes the key findings. Add a brief note about futureresearch directions, such as studying other varieties, testing different concentrations of CO2, or evaluating the success of acclimatization after micro-breeding.
Response: Thank you for the suggestion. We have revised the conclusion (L731-735) to include future research directions (In future research, elevating CO2 concentration and increasing air exchange frequency in culture vessels are expected to further enhance plantlets’ photosynthetic carbon assimilation capacity, thereby promoting robust growth, and improving overall seedlings quality).
- Overall, the English is good, but some sentences are too long or difficult to compose. Tip:Subtract them for minor grammatical improvements and sentence structure optimization. For example: The original: «However, these methods rely on the use of sealed containers and sugar-containing media, which leads to high production costs, low rooting success rates, and low graftsurvival». Improved: «However, these methods rely on the use of sealed containers and sugar-containing media, resulting in high production costs, low rooting success rates, and poor graftsurvival».
Response: Thank you for the helpful suggestion regarding sentence clarity. We have revised several sentences throughout the manuscript to improve readability and optimize sentence structure, particularly by shortening overly long or complex constructions. Specifically, modifications have been made in the following sections: Lines 103–105, 129–131, 269–272, 399–401, 422–425, 576-79, 587-592, and 596–599.

Round 2
Reviewer 1 Report
Comments and Suggestions for Authors
The authors have made all the suggested corrections. The article can now be accepted in its current form.